# Salt Tolerance Diversity in Citrus Rootstocks Agrees with Genotypic Diversity at the *LCl-6* Quantitative Trait Locus

**DOI:** 10.3390/genes16060683

**Published:** 2025-05-30

**Authors:** Maria J. Asins, M. Verónica Raga, Maria R. Romero-Aranda, Emilio Jaime-Fernández, Emilio A. Carbonell, Andres Belver

**Affiliations:** 1Instituto Valenciano de Investigaciones Agrarias (IVIA), 46113 Moncada, Spain; 2Integrative Biology for Plant Stress Group, La Mayora Institute of Subtropical and Mediterranean Horticulture, IHSM-CSIC-UMA, 29010 Malaga, Spain; rromero@eelm.csic.es (M.R.R.-A.); emilio.j.f@csic.es (E.J.-F.); 3Department of Stress, Development and Signaling of Plants, Estación Experimental del Zaidín, Consejo Superior de Investigaciones Científicas (CSIC), C/Prof. Albareda 1, 18008 Granada, Spain; andres.belver@eez.csic.es

**Keywords:** rootstock breeding, germplasm enhancement, *Poncirus*, QTL, NPF5.9, CHX20, Cl^−^ homeostasis, Na^+^ homeostasis

## Abstract

Background/Objective: Salinity is a growing problem affecting a large portion of global agricultural land, particularly in areas where water resources are scarce. The objective of this study was to provide physiological and molecular information on salt-tolerant citrus rootstocks to mitigate the detrimental effects of salinity on citriculture. Methods: Ten accessions belonging to eight *Citrus* species and four to *Poncirus trifoliata* Raf. were tested for salinity tolerance (0 and 15 mM NaCl for 1 year) in terms of vegetative and Cl^−^ tissue distribution traits. In addition, most accessions were evaluated for leaf Na^+^ and other cations. Results: All salt tolerant accessions tended to restrict the leaf Cl^−^ content, although in a lower degree than the Cleopatra mandarin. However, differences in their ability to restrict leaf [Na^+^] were evident, contributing to a classification of trifoliate and sour orange accessions that matched their genotypic grouping based on allele sharing at a marker targeting candidate gene coding for the NPF5.9 transporter within *LCL-6* quantitative trait locus. Conclusions: Our markers targeting *LCl-6* candidate genes coding for NPF5.9, PIP2.1, and CHX20 (citrus GmSALT3 ortholog) could be efficient tools for managing the detected salt tolerance diversity in terms of both Cl^−^ and Na^+^ homeostasis in rootstock breeding programs derived from these species, in addition to *Citrus reshni*.

## 1. Introduction

Salinity in agriculture is a growing issue that negatively affects crop production and soil health. Soil salinity occurs when there is an accumulation of soluble salts, often due to poor irrigation practices, excessive water evaporation, or the use of saline water sources, including desalinized seawater [1,2]. It affects 2.1% of dry agricultural land and up to 19.5% of irrigated land, adding up to a third of world food production, and contributes to a reduction of up to 50% in agricultural yields in some affected areas [3]. Salinity causes tissue burning and leaf abscission in citrus plants, which are classified among the most salt-sensitive tree crops [4]. In fact, tree growth and fruit yield of citrus species are impaired at a soil salinity of approximately 2 dS/m soil saturation, without the concomitant expression of leaf symptoms [5,6]. Additionally, salinity predisposes citrus tree roots to attack by *Phytophthora* [7], nematodes [8], and bacterial pathogens [9]. On the other hand, moderate salinity has been suggested as a management strategy to minimize psyllid settlement and reproduction, thus reducing the spread of Huanglongbing (Citrus Greening Disease), which is caused by the bacterium *Candidatus Liberibacter* spp. and transmitted by the insect in citrus grown in semiarid and arid areas [10]. Obtaining salt-tolerant plants is an effective practice for mitigating the effects of salinity and climate change on citriculture production and orchard sustainability.

Since citrus varieties of sweet oranges, mandarins, grapefruits, pummelos, and lemons are always propagated by bud-grafting onto a seedling rootstock in order to obtain a uniform orchard tolerant of soil pathogens and well-adapted to the local edaphoclimatic conditions, salt tolerance is often a target trait in breeding programs for citrus rootstocks [11]. Citrus germplasm is highly diverse and includes several species showing a certain degree of salt tolerance such as Cleopatra and Sunki mandarins, sour oranges, Volkamer and Rough lemons, Rangpur lime, and Alemow [12,13,14,15,16,17,18,19,20]. The inheritance of the salt tolerance of Cleopatra mandarin (*C. reshni*) was repeatedly studied in a segregating population derived from the cross between *C. reshni* and *P. trifoliata* [18,21]. These genetic analyses of salt tolerance allowed the detection of a repeatable quantitative trait locus (QTL) on chromosome 6 (*LCl-6*), where one of the two *C. reshni* haplotypes in the root was responsible for leaf Cl^−^ exclusion, and provided markers targeting candidate genes that would facilitate the selection of favorable alleles in citrus breeding programs. These candidate genes were supported by their mRNA expression and protein function because they encode transporters potentially involved in Cl^−^ exclusion, such as several members of the Nitrate Transporter 1/Peptide Transporter Family (NPF) [22], namely NPF5.9, NPF5.12, NPF8.1, and NPF8.2, as well as a Major Facilitator Superfamily (MFS) protein, an ABC Transporter G family 17-related protein (ABCG), and a PIP2-1-related aquaporin [21]. Notably, no citrus homologs of well-characterized Cl^−^ or NO_3_^−^ transporters previously implicated in salt stress responses were identified within this QTL, such as some NPF members, NRT2 (NitRate transporter 2), SLAH (SLow-type Anion channel Homologs), SLAC (SLow Anion Channel-associated), CCC (Cation-Cl– Cotransporter), ICln (Ionic Conductance L-type Nonspecific), ALMT (ALuminum-activated Malate Transporter), CLC (ChLoride Channel), or NAXT (Na+/H+ eXchanger Transporter) [23,24,25,26,27]. However, some candidate genes within *LCl-6* corresponded to orthologs with known functions in other species. In Arabidopsis, NPF5.9 is involved in iron homeostasis [28], while NPF5.12, along with NPF5.11 and NPF5.16, mediates vacuolar nitrate efflux at the tonoplast [29]. In rice, NPF8.1 encodes a high-affinity di-/tri-peptide transporter whose expression is induced by nitrogen deficiency, salt, drought, and ABA. Its overexpression enhances stress tolerance, suggesting its role in organic nitrogen redistribution and adaptation [30]. In citrus, MFS and ABCG transporters have been identified and shown to be differentially regulated during biotic stress (e.g., citrus canker), indicating their possible roles in ion transport and abiotic stress responses [31]. Aquaporins of the PIP2-1 subfamily, which are widely associated with water and small solute transport, are also known to participate in abiotic stress responses [32,33]. Another candidate gene within *LCl-6* was a Cation/Proton Antiporter CHX20 (Ciclev10011060), whose root expression was significantly higher in the salt-tolerant hybrid than in the salt-sensitive hybrid [21]. This gene is orthologous to GmSALT3 (GmCHX1) in soybean, which underlies the well-characterized *Ncl* QTL responsible for Cl^−^ exclusion/inclusion [34,35,36,37]. In soybean, the *Ncl* locus regulates Cl^−^, Na^+^, and K^+^ homeostasis [36]; however, the underlying physiological mechanisms remain unclear [37].

Crop domestication and improvement ultimately result in the isolation of genetic diversity valuable to agriculture from wild species. This reduction in diversity affects potentially valuable genetic variants and their associated phenotypes [38]. Moreover, as these authors stated, information on allelic diversity and its phenotypic effects is an essential requirement for genome editing techniques that are now gaining ground for the improvement of crops. Salt tolerance has hardly been studied within citrus species, assuming, for example, that all sour orange (*Citrus aurantium*) accessions are salt-tolerant and all trifoliate oranges (*P. trifoliata*) are salt-sensitive. Variability in the response to salinity has been observed among *P. trifoliata* accessions from the People’s Republic of China [16], which is important because *P. trifoliata* is the main source of resistance genes in breeding programs for citrus rootstocks worldwide. Moreover, *P. trifoliata* shows a kind of resistance to *Trioza erytreae*, the psyllid vector in the Mediterranean Basin transmitting the bacteria causing the devasting citrus disease Huanglongbing [39] and against Huanglongbing itself [40].

Two objectives were pursued in this study: to study the variability in the response to salinity in a core collection of citrus rootstocks and to evaluate the agreement between this phenotypic variability and the genotypic variability at each candidate gene within *LCl-6* (genes coding for NPF5.9, NPF5.12, MFS, ABCG, PIP2.1, and CHX20; [21]) to provide further evidence supporting their contribution to salt tolerance in citrus.

## 2. Materials and Methods

### 2.1. Plant Materials

A total of 14 accessions, 10 from *Citrus* species and 4 from *P. trifoliata* Raf., most of them used as citrus rootstocks, constituted the whole collection of the present salt tolerance study (Table 1). Additional variables (cation leaf contents) were measured in all plants of a part of this collection, named the within-species collection, which excluded *C. sunki* and the lemon/lime group of species. The accessions belonging to *C. aurantium* and *P. trifoliata* were not chosen at random but rather represented a maximum of the molecular variability observed in previously reported germplasm studies regarding *Citrus Tristeza Virus* (CTV) resistance [41,42]. Thus, sour orange Afin Verna is CTV resistant, whereas sour oranges Clementina and Guo-Kuo Cheng accumulate CTV virions after 30 months of CTV challenge [41]. Since trifoliate accessions Flying Dragon and Rich were polymorphic for the candidate gene NPF5.9 [21], four genetically diverse *P. trifoliata* accessions were included, all of which were CTV-resistant [42].

Seeds from all accessions were obtained from the Citrus Germplasm Bank at I. V. I. A. (IVIA references in Table 1). Seedlings were grown in a nursery until the leaves were large enough for DNA extraction, as reported previously [43] with some modifications. At least ten seedlings per entry were genotyped for markers to discard zygotic seedlings. Six to eight nucellar seedlings per accession were sent and grown at the Estación Experimental La Mayora (IHSM-CSIC) in Malaga, Spain, where the salt tolerance experiment was conducted.

### 2.2. Salt Tolerance Experiment and Trait Evaluation

A minimum of three plants from 6/8 repetitions of each accession were randomly selected to be subjected to control and salinity treatments over a period of a year from 18 May 2023, to 14 May 2024, in a greenhouse under natural light conditions with no temperature control. Plants were grown under natural greenhouse conditions in pots (4 L) using a vermiculite substrate. The nutrient solution (electrical conductivity: 1.68, pH: 7.13) contained the following concentrations of macronutrients (in mM): NO_3_^−^ 8.45; H_2_PO_4_^−^ 0.74; SO_4_^2−^ 1.84; K^+^ 6.08; Ca^2+^ 4.25; and Mg^2+^ 1.33; in addition to the following concentrations of micronutrients (in µM): Fe^2+^ 66, Mn^2+^ 1.33; Zn^2+^ 2.3; B 17, and Cu^2+^ 2.3, supplemented with either 0 mM NaCl for the control (1.67 dS m^−1^) or 15 mM NaCl for the saline treatment (4.67 dS m^−1^) until completion of the experiment (12 months). Plants were automatically watered using 2.5 L h^−1^ drippers three times a week, on alternate days, receiving 200 mL at each irrigation event.

The whole collection was evaluated for leaf symptoms, vegetative traits, and Cl^−^ concentration in the leaves (L_Cl) and roots (R_Cl). To characterize the organ distribution of Cl^−^, the function ([Cl]root − [Cl]leaf)/[Cl]root was calculated and used as an additional trait (ClR-ClL/ClR) at both salinity levels.

Vegetative traits were plant height (H), stem diameter (4 to 5 cm above substrate) (SD), number of leaves (LN), and number of branches (BN) per plant, at two dates, 7 and 11 months from the beginning of the experiment denoted by suffixes 7 and 11, respectively, and the mean difference between dates for SD (dSD) and LN (dLN). At the end of the experiment, all leaves from each plant were collected and dried to obtain their dry weight (LDW) in grams and the same for the stem (SDW). The roots of each plant were washed thoroughly with tap water and dried. Total root dry weight (RDW)) was estimated for each plant in grams. Total plant dry weight (tPDW) was obtained by summing LDW, RDW, and SDW. The root-to-total plant dry weight ratio (RDW/tPDW) was also considered.

Foliar and root concentrations of Cl^−^ (mg L^−1^) were evaluated using a chloride analyzer (Model 926, Sherwood Scientific, Cambridge, UK) and the methodology described by Gilliam [44].

The leaf Na, Ca, K, Fe, Mg, S, Si, P, and B contents were determined as a percentage of dry weight (ppm) using a Varian 720-E inductively coupled plasma-optical emission spectrometer (ICP-OES; Scientific Instrumentation Service, EEZ, CSIC, Granada, Spain). These elements were measured only in the within-species collection.

### 2.3. Candidate Gene Genotyping

Most primers corresponding to sequence-characterized amplified region (SCAR) markers to genotype accessions for candidate genes at the *LCl_6* QTL were developed and mapped previously [21]. Primers for the CHX20 candidate (Appendix A) (Ciclev10011060m at scaffold_6:13305253..13310801, coding for a CATION/H^+^ antiporter 20) were used to map this gene within *LCl-6* following the same methodology described previously [21], and to genotype all accessions in the present study of molecular diversity. This gene is a citrus ortholog of GmSALT3, a gene crucial for salt tolerance in soybeans, that acts primarily through Na^+^ exclusion and the regulation of Cl^−^ transport [37].

Genomic DNA was extracted from the leaf tissue of each plant. Polymerase chain reaction (PCR) conditions were specific to each marker, and the resulting product was analyzed by electrophoresis in 10% DNA sequencing-type polyacrylamide gels and revealed by silver staining. All procedures were described previously [43].

### 2.4. Statistical Analysis

Two-way fixed-effects analysis of variance was used to study the genotype (G) and salinity (E) effects, as well as the G × E interactions of the evaluated traits. Additionally, the genotypic effects were analyzed for each treatment. Mean comparisons were performed using the test by Di Rienzo, Guzmán, and Casanoves (DGC) [45], allowing the means to be classified into disjunct groups, and the correction for multiple comparisons of Benjamini and Hochberg [46] was applied. This type of disjoint classification for each quantitative trait was used to build dendrograms using the Jaccard distance and average distance as a clustering procedure for the accessions. Similarly, the genotype data (allele composition) of all accessions for relevant candidate genes within *LCl-6* (NPF5.9, NPF5.12, MFS, ABCG, PIP2.1, and CHX20) and microsatellite marker CR23 [21] were used to obtain dendrograms of accessions using the same procedure as before. The Infostat [47] statistical package was used for data analysis and graphs.

## 3. Results

Although all selected accessions in the collection of rootstocks (Table 1) are apomictic, yielding polyembryonic seeds, the percentage of zygotic seedlings (from sexual origin) was zero only in sour orange accessions, Cleopatra and Sunki mandarins, Alemow, and Volkamer and Rough lemons. A noticeable difference in the percentage of zygotic seedlings was observed among trifoliate oranges, ranging from 9% for Benecke up to 36% for Pomeroy. Discarding zygotic plants by screening for molecular marker segregation provides reliability for this citrus germplasm study. 

### 3.1. Salinity Effects on Symptomatology and Vegetative Development

*Poncirus* accessions showed clear differences in salt-related symptoms (Appendix A). Thus, Tr-236 (Rich) was very sensitive to salinity, while Tr-376 (Benecke) was the least damaged trifoliate accession. The group of limes and lemon rootstocks (Ma-288, Ra-334, Ru-333, and Vo-432) under saline conditions showed only some necrotic areas in the lowest-level leaves. *C. aurantium* accessions presented fewer salt injuries than those in the previous group. Only Au-141 (Clementina) plants developed brownish leaves.

Statistical analysis of vegetative traits in the entire collection showed significant genotypic effects (G) for all traits (Table 2). Salinity effects (E) were generally significant, with a few exceptions. Only variation in the number of leaves per plant (dLN) showed significant G × E in the sense that most accessions did not show important differences between the two treatments, whereas there was a clear difference in the sweet orange accession Si-011, which lost leaves under salinity (Figure 1). Similarly, the development of new leaves in the sour orange accessions under salinity was lower than that under the control, particularly in Au-141 (Clementina). Regarding trifoliata orange accessions, Tr-537 clearly developed fewer leaves under saline conditions than under control conditions.

Leaf dry weight (LDW) under salinity decreased in Au-130 and Si-011 and remained constant in the rest, except for Rangpur lime, Ra-334, which increased. The development of the stem diameter (dSD) was rarely affected by salinity, except for Au-183 (Guo-Kuo-Cheng), whose stem diameter grew wider under salinity than in control. Root dry weight (RDW) decreased under salinity, particularly in Ru-333, Au-138, Si-011, Tr-236, and Tr-537. No RDW changes were observed in Vo-432, Ma-288, Ra-334, Au-141, Au-183, Su-239, or Tr-374 (Pomeroy). Taking into account the dry weight of the whole plant (tPDW) as a measure of salt tolerance, this parameter clearly decreased in Tr-236, Tr-537, Si-011, Au-130, and Ru-333 (Figure 1), and the RDW/tPDW ratio decreased in Ru-333, Tr-376, Tr-236, Ra-334, and Tr-537 (Appendix A).

To better visualize the variability between and within species, the results of the mean comparisons were used to classify the accessions for each trait/treatment combination and both treatments together (Appendix A), providing quite different groupings. Notably, without considering the leaf cation contents, the trifoliate accessions grouped together only when phenotypic data from both treatments were used for dendrogram construction (Appendix A). In the case of the sour oranges, they grouped together, with Volkamer lemon, only under salinity (Appendix A).

### 3.2. Ion Homeostasis

The analysis of variance for leaf ion contents (Table 2 and Table 3) revealed differential behavior among accessions in the two treatments (significant G × E) for R_Cl, L_Cl, L_B, L_Mg, and L_Na. Thus, Re-385 and Ra-334 maintained similar leaf Cl^−^ contents under control and salinity conditions, whereas Tr-236 clearly accumulated Cl^−^ in its leaves (Figure 2). The root Cl^−^ content (R_Cl) was similar between treatments in Vo-432 and Au-183, whereas it was highly increased in Tr-537. The latter explains why Tr-537 maintains the root to leaf Cl^−^ distribution (RCl-LCl/RCl) similarly between treatments, making the root a type of Cl^−^ reservoir (likely preventing its growth, Figure 1), contrary to Tr-236, which seems to send most Cl^−^ to the leaf, leading to plant death (Appendix A).

Interestingly, clear within-species variability was observed in leaf Na^+^ content (L_Na, Figure 2). Thus, Au-183 (Guo-Kuo-Cheng) behaved as a Na^+^ leaf excluder, similar to most trifoliate accessions (Tr-236, Tr374, and Tr-376). However, Tr-537 (Flying Dragon) behaved as a Na^+^ leaf includer, such as Au-130, Re-385, Au-141, and Si-011. The latter two accessions did not tolerate much Na^+^ leaf content because Au-141 had some brownish leaves (Appendix A), and Si-011 lost some leaves under saline conditions (dLN in Figure 1).

Other differential behaviors within species were observed in Tr-537 in comparison to the other trifoliate accessions for L_B, L_Mg, and L_S, and in Au-130 (Afin Verna) for L_Fe and L_Mg with respect to the other sour orange accessions.

Similarly, as with vegetative traits, the accessions of the within-species collection could be classified according to the results of the mean comparison grouping regarding leaf cation contents (Appendix A). In this case, dendrograms representing accession behaviors under control and salinity conditions regarding vegetative and ion traits were very similar. These dendrograms are shown in Figure 3A and Figure 3B, respectively. The main difference corresponded to the grouping of sour orange accessions, particularly Au-130 (Afin Verna).

### 3.3. Candidate Gene Polymorphisms at LCl_6 QTL

Since no recombinants were detected when genotyping the reference segregating population for the SCAR markers targeting MFS and CHX20, genes coding for MFS (Ciclev10011745) and CHX20 (Ciclev10011060) were located exactly at the same genetic position in the *C. reshni* map.

Molecular polymorphisms at markers targeting candidate genes within the *LCl-6* QTL (coding genes for NPF5.9, PIP2.1, MFS, CHX20, ABCG, and NPF5.12 transporters) were studied in the entire collection of accessions (Appendix A). Classification of accessions with regard to their allele sharing (Appendix A) was used to build dendrograms based on all of them together, each candidate gene separately, and the microsatellite marker CR23, which was located within the QTL (Figure 4 and Appendix A). No general agreement between phenotypic and genotypic dendrograms was observed when using the whole collection (and no leaf cation content trait was considered), except for the group of trifoliate oranges (Appendix A). However, the phenotypical classification of accessions in the within-species collection (Figure 3) was similar to that obtained using the joint genotype for all candidates (Figure 4A), and in particular, using only the genotypic classification for NPF5.9 (Figure 4B). The classification of the genotypes at this candidate gene agreed with the phenotypic classification of trifoliate accessions and the classification of accessions regarding leaf Na^+^ content under salinity (Figure 2). Thus, Tr-537 is a leaf Na^+^ includer (in addition to a Cl^−^ root includer) in comparison to the other trifoliate accessions, and Au-183 is a leaf Na^+^ excluder in comparison to the other sour orange accessions that group together for NPF5.9 and are leaf Na^+^ includers (Figure 4). This separation between sour orange accessions was also shown by dendrograms using CHX20 or MFS genotypes.

## 4. Discussion

### 4.1. Diversity in Salt Response Among Citrus Species

Citrus rootstocks conferring salt tolerance showed a remarkable diversity in vegetative development (Figure 1 and Appendix A). Thus, Rough lemon, Rangpur lime and Alemow are the most vigorous plants under salinity (tPDW, SDW in Figure 1, Appendix A). They, together with Volkamer lemon and the sour orange accessions, have larger root systems than Sunki and Cleopatra mandarins. Notably, the root biomass of Vo-432 under salinity was slightly larger than that under control conditions (Figure 1), as observed in the salt-tolerant citrus rootstock US-942, suggesting they may share a similar mechanism against moderate salinity [48].

Leaf Cl^−^ accumulation under salinity, a feature commonly related to salt sensitivity in citrus [49], was high only in two trifoliate accessions, Tr-236 which showed signs of leaf damage, and Tr-374, but did not differ for the remaining accessions and species. In fact, the range of means for L_Cl under salinity (Figure 2) was much narrower than that observed for the segregant population derived from the cross between Cleopatra mandarin and *P. trifoliata* [21], suggesting that most of them (except Tr-236 and Tr-374) exclude Cl^−^ from the leaves; i.e., are salt tolerant. It is noteworthy that some accessions (Cleopatra, Volkamer and rough lemons), showed a higher root [Cl^−^] than the others under control conditions. Among them, Rough lemon (Ru-333) showed a great root development. Trees on rough lemon are known to be very drought tolerant [50,51]. As chloride is an essential micronutrient with proposed regulatory roles in photosynthesis, transpiration, fertilization, nutrition, and growth [23] at low concentration, it could increase water and nitrogen efficiencies and mitigate the effects of water deficit [52], which may be the case with Rough lemon.

### 4.2. Diversity in Salt Response Among Trifoliate Orange Accessions Matches the Genotype Diversity at NPF5.9 and PIP2.1

*P. trifoliata* germplasm is a rich source of resistance alleles against important citrus diseases such as Tristeza [42], root rot [53], and Huanglongbing [39,40]. However, a high percentage of zygotic seedlings in trifoliate orange accessions is a major handicap when studying this germplasm, as it increases experimental error due to the resulting genetic heterogeneity derived from zygotic seedlings. In contrast to *Citrus* and *Fortunella*, where nucellar embryony has been shown to be caused by a mutated gene with full penetrance, CitRWP [54,55] or its ortholog FhRWP [56], the causal gene(s) in *Poncirus* remains unknown [56,57]. The variation in the percentage of zygotic seedlings observed in polyembryonic trifoliate oranges, ranging from 9% in Benecke to 36% in Pomeroy suggests the absence of a single fully dominant gene as the responsible gene for this trait (percentage of zygotic seedlings) in *Poncirus*.

The dendrograms in Figure 3 show the same classification of the trifoliate accessions under control and salinity conditions, but their explanation relates to different traits. In both cases, Tr-537 stands out from the other three accessions. Under control conditions, this differentiation is due to the low values of Tr-537 for dSD and L_B, while under salinity, it is explained by the high values for R_Cl and L_Na, and low for L_S (Appendix A). Boron is a critical micronutrient for plant growth and can alleviate aluminum toxicity in trifoliate orange by inhibiting cell wall deposition and promoting vacuole compartmentation [58]. Aluminum is only phytotoxic under acidic conditions, but approximately 49% of the world’s arable land is affected by acidic soils [59].

The observed salt injury (Appendix A), large leaf Cl^−^ accumulation, low value of RCl-LCl/RCl (Figure 2), and large decrease in tPDW (Figure 1) clearly point to Tr-236 as the most salt-sensitive accession in the whole collection. On the other hand, the lack of salt injury and the low leaf Cl^−^ content under salinity of Tr-376 (Figure 2) suggest that Benecke has some degree of salt tolerance.

The phenotypic discrimination of Flying Dragon (Tr-537) from the other accessions in the salt tolerance experiment (Figure 3) agrees with its genotypic differentiation for the SCAR NPF5.9 (Figure 4B and Appendix A) and its banding pattern for PIP2.1 (Appendix A). Rubiduox, which has the same genome sequence as Flying Dragon [60], has been considered a salt-tolerant trifoliate accession [17]; however, Flying Dragon (Tr-537) has shown a significant decrease in values from control to saline conditions for several vegetative traits (dLN, RDW, tPDW), suggesting a certain sensitivity that must be related to its lack of ability to exclude Na^+^ from leaves, since it seems to exclude Cl^−^ from leaves by retaining it in the root (Figure 2). This ability to accumulate Cl^−^ in the root, in contrast to Tr-236, could explain why Flying Dragon was less affected by water deficit than other trifoliate accessions in another experiment [61]. Thus, well-watered tobacco plants increased the efficiency of water, nitrogen, and CO_2_ use at low [Cl^−^] (1-5mM) and improved their ability to withstand drought [52]. The detection of salt tolerance in Benecke, the trifoliate orange with the lowest percentage of zygotic seedlings (9%), provides useful information for rootstock breeding purposes and the sustainability of citriculture in regions affected by salinity.

Therefore, both Cl^−^ and Na^+^ homeostasis appear to be relevant for plant salt tolerance in trifoliate orange accessions, as previously reported [16]. Is Na^+^ homeostasis also controlled, at least in part, by the *LCl-6* QTL? Not all salt tolerance experiments constructed using the same reference population derived from the cross between *C. reshni* and *P. trifoliata* yielded significant QTL in its genomic region for leaf Na^+^ content [18,21], in contrast to leaf Cl^−^ content. Indeed, these inheritance studies (using grafted and non-grafted plants) also showed significant effects for K, Ca, and Fe in the genomic region of *LCl-6*, but not in all experiments, suggesting they have minor effects compared to the effect on leaf Cl^−^ exclusion. In *Arabidopsis*, NPF5.9 is a multifunctional transporter involved in both nitrate and Fe transport. Its role in Fe homeostasis is independent of its nitrate transport function, highlighting the complexity of nutrient regulation in plants [28]. In addition to nitrate and chloride, NPF proteins have been shown to transport a broad variety of substrates [27]. PIP2.1 is also a likely candidate that could explain the phenotypic divergence of the drought tolerant Flying Dragon from the other trifoliate orange accessions. In *Arabidopsis*, PIP2.1 transports a wide range of substrates including water, monovalent cations, and hydrogen peroxide, participating in the regulation of the stress response [62,63]. The substrates of NPF5.9 and PIP2.1 in citrus are currently under investigation.

The joint homeostasis of Cl^−^ and Na^+^ by the plant reminds the functioning of GmSALT3 in soybean [37]. Candidate CHX20 (Ciclev10011060) is an orthologue of GmSALT3 in citrus. Its genetic position is the same as that of the MFS candidate, where the QTL (*LCl-6*) peak is located [21]. Nevertheless, primers used to genotype the collection for CHX20 did not show any polymorphism among the trifoliate accessions, while the NPF5.9 and PIP2.1 SCARs did (Appendix A). Obviously, the lack of polymorphism detection does not mean that there are no allelic differences among accessions at CHX20 but does not provide evidence to support it as a causal gene, in contrast to NPF5.9 and PIP2.1.

### 4.3. Diversity in Salt Response Among Sour Orange Accessions Is Related to Na^+^ Homeostasis and Is Consistent with Their Genotype Classification at NPF5.9 and CHX20

Among the sour orange accessions selected for this salt tolerance study because of their diversity regarding *Citrus Tristeza Virus* (CTV) resistance markers and ability to accumulate CTV virions [41], Au-130, the only CTV-resistant accession within this group, is phenotypically the most distant one, especially under control conditions (Figure 3A). The main traits involved are RDW and SD under control conditions and RDW/tPDW, L_K and L_Mg under both conditions (Appendix A). Among the candidate genes, this separation of Au-130 is only observed for the genotypic classification at NPF5.12 (Figure 4E).

According to our results on symptoms, the least salt-tolerant sour orange accession is Au-141 (cultivar Clementina), but no significant difference among sour orange accessions was observed regarding leaf or root Cl^−^ contents or Cl^−^ distribution between root and leaves (Figure 2, Appendix A). The main significant difference concerns their ability to exclude Na^+^ from leaves. Thus, Figure 2 shows that Au-183 (Guo-Kuo-Cheng cultivar), similarly to most trifoliate accessions, is able to exclude Na^+^ from leaves while sweet orange, Cleopatra, and the other sour orange accessions are not. A similar classification was obtained using the genotype of accessions at NPF5.9 as a grouping criterion (Figure 4B) where Au-183 (Na^+^-excluder) is far apart from the group formed by Au-141, Au-130, and Si-011, and Re-385 (all Na^+^-includers). Classification of genotypes at CHX20 also discriminates the Guo-Kuo-Cheng cultivar (very close to Cleopatra) from the Clementina and Afin-Verna sour cultivars (Figure 4G). The Guo-Kuo-Cheng cultivar was selected also as a salt-tolerant rootstock in previous studies [14,16]. Since it has been shown to accumulate CTV virions, in contrast to most sour accessions, including Afin-Verna [41], it could prevent sweet orange tree decline by Tristeza disease when used as a rootstock instead of most sour accessions, which is something to be tested.

## 5. Conclusions

The response to salinity is diverse within both *C. aurantium* and *P. trifoliata* species, which can enrich our panel of citrus rootstocks and their breeding programs worldwide. Moreover, this diversity is consistent with the genotype diversity detected at the NPF5.9 locus within the salinity tolerance QTL *LCl-6* [21] and supports its relevance as a candidate gene for salt tolerance. Our markers targeting *LCl-6* candidate genes encoding NPF5.9, PIP2.1, and CHX20 (citrus GmSALT3 ortholog) could be efficient tools to manage salt tolerance diversity in terms of both Cl^−^ and Na^+^ homeostasis, in rootstock breeding programs derived from these species, in addition to *C. reshni*.

## Figures and Tables

**Figure 1 genes-16-00683-f001:**
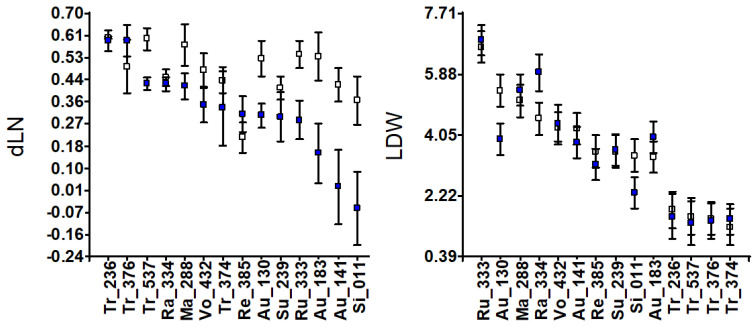
Accession means and standard deviations for vegetative traits: development of new leaves (dLN) and stem diameter (dSD); leaf, stem and root dry weights (LDW, SDW and RDW); and total plant dry weight (tPDW) in grams. White and blue squares indicate control and salinity conditions, respectively.

**Figure 2 genes-16-00683-f002:**
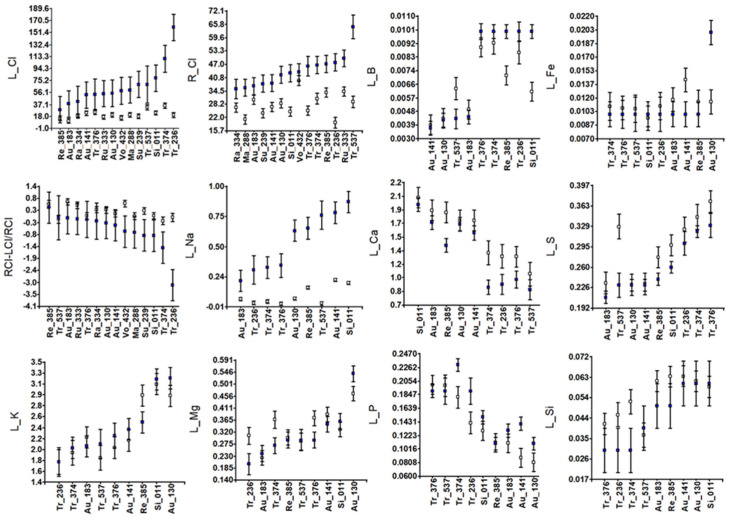
Accession means and standard deviations for ion traits: leaf and root [Cl^−^] in mg L^−1^ (L_Cl and R_Cl), root to leaf Cl^−^ distribution (RCl-LCl/RCl), and leaf concentrations of relevant cations (L_Na, L_K, L_Mg, L_B, L_Fe, L_Ca, L_S, L_P, and L_Si) in ppm. White and blue squares indicate control and salinity conditions, respectively.

**Figure 3 genes-16-00683-f003:**
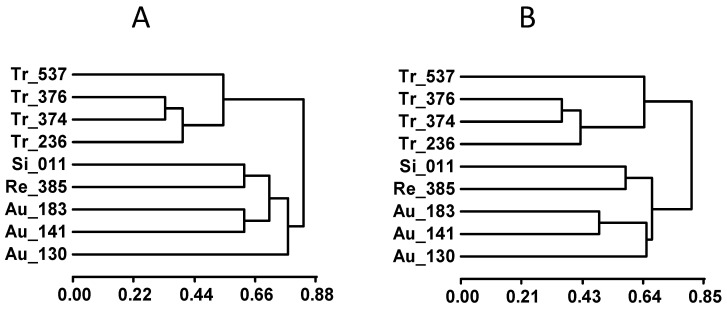
Phenotypic dendrograms of accessions from the within-species collection using their evaluation for all traits under control (**A**) and salinity (**B**) conditions.

**Figure 4 genes-16-00683-f004:**
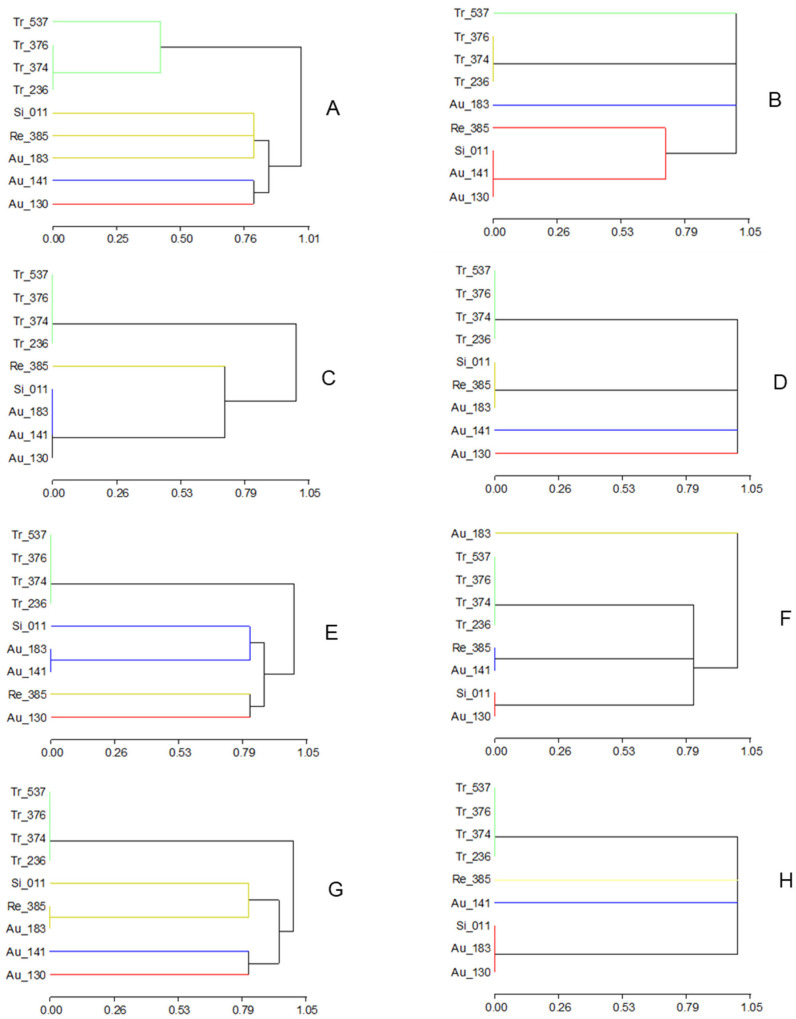
Genotypic dendrograms of accessions from the within-species collection using the allele composition at all candidate genes (**A**), or at each one separately: NPF5.9 (**B**), ABCG (**C**), PIP2.1 (**D**), NPF5.12 (**E**), MFS (**F**) and CHX20 (**G**), and at microsatellite CR23 (**H**).

**Table 1 genes-16-00683-t001:** List of accessions of the rootstock collection ordered by species. A code for each accession is provided where the number corresponds to its reference in the Citrus Germplasm Bank at IVIA.

Species	Common Name	Code
*C. aurantium* L.	Sour orange Afin Verna	Au-130
*C. aurantium* L.	Sour orange Clementina	Au-141
*C. aurantium* L.	Sour orange Guo Kuo Cheng	Au-183
*C. jambhiri* Lush.	Rough lemon	Ru-333
*C. limonia* Osbeck	Rangpur lime	Ra-334
*C. reshni* Hort. Ex Tan.	Cleopatra mandarin	Re-385
*C. sinensis* (L.) Osb.	Sweet orange Pineaple	Si-011
*C. sunki* (Hayata) hort ex. Tanaka	Sunki mandarin	Su-239
*C. volkameriana* Te. & Pasq.	Volkamer lemon	Vo-432
*C. macrophylla* Wester.	Alemow	Ma-288
*P. trifoliata* (L.) Raf.	trifoliate orange Rich	Tr-236
*P. trifoliata* (L.) Raf.	trifoliate orange Pomeroy	Tr-374
*P. trifoliata* (L.) Raf.	trifoliate orange Benecke	Tr-376
*P. trifoliata* (L.) Raf.	trifoliate orange Flying Dragon	Tr-537

**Table 2 genes-16-00683-t002:** *p*-values of significant effects from genotype (G), salinity treatment (E) and their interaction (G × E) in the analysis of vegetative and Cl^−^ traits in the whole collection. * *p*-value when using the within-species collection. Suffixes _7 and _11 indicate months after the salt treatment in the evaluation.

Trait	G	E	G × E
L_Cl	0.0041	<0.0001	0.0451
ClR-ClL/ClR	0.06/0.0001 *	0.0010	
R_Cl	<0.0001	<0.0001	0.0059
H_7	<0.0001		
SD_7	<0.0001	<0.0001	
LN_7	<0.0001		
SD_11	<0.0001	0.0030	
LN_11	<0.0001	0.0020	
BN_11	0.0027		
dSD	0.0001		
dLN	0.0001	<0.0001	0.0387
SDW	<0.0001	0.0329	
LDW	<0.0001		
RDW	<0.0001	0.0007	
tPDW	<0.0001	0.0072	
RDW/tPDW	<0.0001	<0.0001	

**Table 3 genes-16-00683-t003:** *p*-values of significant effects from genotype (G), salinity treatment (E), and their interaction (G × E) in the analysis of leaf cation traits in the within-species collection.

Trait	G	E	G × E
B	<0.0001	<0.0001	0.0106
Ca	<0.0001	0.0002	
Fe	0.0403		
K	<0.0001		
Mg	<0.0001		0.0479
Na	<0.0001	<0.0001	0.0003
P	<0.0001	0.0018	
S	<0.0001	0.0004	
Si	<0.0001	0.0044	

## Data Availability

The original contributions presented in the study are included in the article/Appendix A. Further inquiries can be directed to the corresponding author.

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
