# Peer review of "Salt Tolerance Diversity in Citrus Rootstocks Agrees with Genotypic Diversity at the *LCl-6* Quantitative Trait Locus"

_genes, 2025, doi:10.3390/genes16060683_

Round 1
Reviewer 1 Report
Comments and Suggestions for Authors
The information developed in this study is valuable in directing breeding of salt tolerant citrus rootstocks.
However, container salinity studies with young, unbudded, seedling rootstocks are only a good indicator of what to pursue/evaluate in more considered container or field studies with budded rootstocks if the focus is the mechanism of the salinity tolerance.
This study could have been strengthened by performing the study with older seedling rootstocks budded to scions. Perhaps budding the 10 different rootstocks to the same salt sensitive scion (and from the same budwood source tree) would not have been horticulturally/commercially ideal combinations for commercial citrus production but it might have yielded more information about the specific mechanisms of the salt tolerance.
In commercial production the rootstocks do not have leaves so ions would not accumulate there. In a budded rootstock it would have been valuable to see if the salt was excluded at root cortex, sequestered in the rootstock xylem parenchyma or was transported across the bud union to the leaves of the scion.
For example a summary of the pistachio rootstock study of Godfrey et. al.: "The UCBI seedling rootstock is more salinity tolerant because it excludes Na+ and Cl- at the root level, stores Na+ in the vacuoles of the root’s cortex, retrieves Na+ from the xylem stream, storing it in the xylem parenchyma and recirculates Cl- in the rootstock trunk phloem. What is interesting is that none of these mechanisms are present in the UCBI parents even though the seedling rootstocks Drakakai and Godfrey used were almost certainly from the original UCBI parents. These results were produced with budded seedling rootstocks. However, there is no physiological reason to think these mechanisms of salinity tolerance would be different in clonally propagated rootstocks."
I am assuming responses to osmotic pressure were not addressed directly, but indirectly through effects on growth.
In conclusion, this study is a valuable contribution for the future improvement of saline tolerant citrus rootstocks. The information above is a suggestion for future experiments.
Godfrey JM, Ferguson L, Sanden BL, Tixier A, Sperling O, Grattan SR, Zwieniecki MA. Sodium interception by xylem parenchyma and chloride recirculation in phloem may augment exclusion in the salt tolerant Pistacia genus: context for salinity studies on tree crops. Tree Physiol. 2019 Aug 1;39(8):1484-1498. doi: 10.1093/treephys/tpz054. PMID: 31095335.
JM Godfrey, L Ferguson, MA Zwieniecki. 2021. Sodium retrieval from sap may permit maintenance of carbohydrate reserves in mature xylem tissues of a salt-tolerant hybrid pistachio rootstock exposed to 100 mm NaCl. Journal of the American Society for Horticultural Science 146 (4), 224-232.
S Zhang, A Quartararo, OK Betz, S Madahhosseini, AS Heringer, T Le, Y Shao, T Caruso, L Ferguson, J Jernstedt, T Wilkop, G Drakakaki. 2021. Root vacuolar sequestration and suberization are prominent responses of Pistacia spp. rootstocks during salinity stress. Plant Direct 5 (5), e00315.
One small question/correction: line 196 "cero"...should this be "zero".
You write well in English; wish I could do as well in your language.
Author Response
Comments 1: However, container salinity studies with young, unbudded, seedling rootstocks are only a good indicator of what to pursue/evaluate in more considered container or field studies with budded rootstocks if the focus is the mechanism of the salinity tolerance.
This study could have been strengthened by performing the study with older seedling rootstocks budded to scions. Perhaps budding the 10 different rootstocks to the same salt sensitive scion (and from the same budwood source tree) would not have been horticulturally/commercially ideal combinations for commercial citrus production but it might have yielded more information about the specific mechanisms of the salt tolerance.
In commercial production the rootstocks do not have leaves so ions would not accumulate there. In a budded rootstock it would have been valuable to see if the salt was excluded at root cortex, sequestered in the rootstock xylem parenchyma or was transported across the bud union to the leaves of the scion.
Response 1: Yes, we agree. A salt tolerance experiment with older seedlings budded to the same salt-sensitive scion could have provided physiological information that is closer to the commercial citriculture conditions, although it would have taken 6 years at least.
As mentioned in the introduction section our germplasm study on salt tolerance was based on previously reported genetic studies [18,21], one of them using the segregating population of rootstocks grafted with the same mandarin cultivar [21]. Therefore, the repeatedly detected salt tolerance LCl-6 QTL, and the candidate genes that includes, must play an important role in the citrus roots to tolerate salinity. Is any of these candidates more relevant than the others to explain the phenotypic diversity in salt response among accessions? This is the main question we try to answer in the present paper using a core collection of citrus rootstocks.
Comment 2: I am assuming responses to osmotic pressure were not addressed directly, but indirectly through effects on growth.
Response 2: Yes, you are right.
Comment 3: In conclusion, this study is a valuable contribution for the future improvement of saline tolerant citrus rootstocks. The information above is a suggestion for future experiments.
Response 3; Yes, future salinity experiments using relevant accessions from present paper, such as Benecke and Guo-Kuo-Cheng are needed to obtain physiological information on the salt tolerance mechanisms involved.
Comment 4: line 196 "cero"...should this be "zero"
Response 4: Yes, we have changed it.
Reviewer 2 Report
Comments and Suggestions for Authors
I believe this manuscript is of a high quality and addresses an interesting topic, particularly in the context of the increasing use of desalinated seawater. This study provides new data that will allow to adapt the global agriculture to a context where the periodic increase in droughts will push to use higher amounts of desalinized seawater. However, some aspects should be clarified.
The authors mention a list of genes that may serve as useful tools for breeding salt-tolerance crops. Among these genes, they described appropiately the role of some of them such as NPF5.9 and CHX20 related with Cl- and Na+ homeostasis. However, I believe the role of the mentioned PIP2;1 may be also important in salinity resistance and is barely discussed in the manuscript. This should be described in more detail.
The whole study focuses on the Cl- and Na+ content in root and leaves. However, the presence of PIP2 ;1 and the fact that this protein is involved in the H2O2 transport makes me think that maybe differences in efficiency in H2O2 transport may have an effect in salinity resistence (Less efficiency less damaged leaves ?). Did de authors measure the H2O2 content in different tissues and varieties ?

Author Response
Comments 1: The authors mention a list of genes that may serve as useful tools for breeding salt-tolerance crops. Among these genes, they described appropiately the role of some of them such as NPF5.9 and CHX20 related with Cl- and Na+ homeostasis. However, I believe the role of the mentioned PIP2;1 may be also important in salinity resistance and is barely discussed in the manuscript. This should be described in more detail.
Response 1: Yes, you are right. Thank you for this insightful comment. We have added several sentences (lines 396-401, in blue) and references regarding the substrates of PIP2.1 in Arabidopsis and its role in the regulation of the stress reponse through oxidative signaling.
You are suggesting a compelling hypothesis, however, we still need to study the substrates of citrus PIP2.1 as well as those of citrus NPF5.9.
Comments 2: The whole study focuses on the Cl- and Na+ content in root and leaves. However, the presence of PIP2 ;1 and the fact that this protein is involved in the H2O2 transport makes me think that maybe differences in efficiency in H2O2 transport may have an effect in salinity resistence (Less efficiency less damaged leaves ?). Did de authors measure the H2O2 content in different tissues and varieties ?
Response 2: No, we did not measure hydrogen peroxide contents in this experiment.